**Data Availability Statement:** All data and script are available from OSF at https://osf.io/g2k6z/.

**Funding:** Camille Grasso was supported by a doctoral fellowship of the French Ministry of Higher Education, Research, and Innovation. This study

# Embodied time: Effect of reading expertise on the spatial representation of past and future

**Camille L. Grasso**[1]*, **Johannes C. Ziegler**[1], **Jennifer T. Coull**[2], **Marie Montant**[1]

**1** CNRS, Laboratoire de Psychologie Cognitive (UMR 7290), Aix-Marseille Université, Marseille, France,
**2** CNRS, Laboratoire de Neurosciences Cognitive (UMR 7291), Aix-Marseille Université, Marseille, France

* Camille.grasso@univ-amu.fr

## Abstract

How do people grasp the abstract concept of time? It has been argued that abstract concepts, such as *future* and *past*, are grounded in sensorimotor experience. When responses to words that refer to the past or the future are either spatially compatible or incompatible with a left-to-right timeline, a space-time congruency effect is observed. In the present study, we investigated whether reading expertise determines the strength of the space-time congruency effect, which would suggest that learning to read and write drives the effect. Using a temporal categorization task, we compared two types of space-time congruency effects, one where spatial incongruency was generated by the location of the stimuli on the screen and one where it was generated by the location of the responses on the keyboard. While the first type of incongruency was visuo-spatial only, the second involved the motor system. Results showed stronger space-time congruency effects for the second type of incongruency (i.e., when the motor system was involved) than for the first type (visuo-spatial). Crucially, reading expertise, as measured by a standardized reading test, predicted the size of the space-time congruency effects. Altogether, these results reinforce the claim that the spatial representation of time is partially mediated by the motor system and partially grounded in spatially-directed movement, such as reading or writing.

## Introduction

If abstract concepts are intangible, how could they be embodied through body-environment interactions? Embodied theories of language claim that concepts are represented through sensorimotor, interoceptive, and emotional networks that are activated when learning these concepts [1–6]. From this theoretical point of view, it is rather easy to conceive how the meaning of concrete words, such as *book*, could be embodied through sensorimotor experience (e.g., since books are usually handled manually, hearing or reading this word supposedly activates hand-related motor system; for some empirical evidence see for example [7, 8]). However, this is more complex for abstract concepts because we cannot physically"grasp" them through our senses, nor easily identify the cognitive experience during development that led to the representation of abstract words [8, 9].

benefited from support of the Institute of
Convergence ILCB (ANR-16-CONV-0002) and the
Excellence Initiative of Aix-Marseille University
A*MIDEX.

**Competing interests:** The authors have declared
that no competing interests exist.

Consider for instance the abstract concept of time. Although we can feel and conceptualize it, time is an elusive construct devoid of any concrete phenomenology [10, 11]. There is no physical stimulus and no perceptual organ dedicated to time. In the absence of direct sensory experience of past and future, how could the brain construct a mental representation of time? One solution that has been proposed in the literature [12–16] is that time is represented spatially, flowing linearly from one position in space to another. Thus, temporal cognition is thought to reuse neural structures devoted to spatial cognition.

This close link between space and time was theorized under the name of the mental timeline (for reviews see: [15–18]). The mental timeline (MTL) refers to the phenomenon whereby individuals represent the temporal content of verbal stimuli (e.g., past and future) in two spatial axes that are centered on the body: for example, back-to-front (i.e., past in the back and future in the front, e.g., [19–22]); and left-to-right (i.e., past on the left and future on the right, e.g., [1, 13, 23, 24]). When participants make a temporal judgement about a picture for example (e.g., decide whether the picture describes a past or future event), they give faster left-hand responses for past events and faster right-hand responses for future events [25]. In psycholinguistics studies, empirical evidence for the MTL has come from experiments that manipulated congruency between the temporal content of verbal stimuli and the spatial position of the required response, resulting in a space-time congruency effect [17, 18, 26, 27]. In various word or sentence processing tasks investigating the left-to-right MTL, left-hand responses are typically faster than right-hand ones for past-tense stimuli and right-hand responses are faster than left-hand ones for future-tense stimuli, for word stimuli [e.g., 1, 21, 28], and for sentence stimuli [e.g., 29–31]. Interestingly, Li et al. [32] showed that when people operate in their second language, bilinguals unconsciously retrieve irrelevant native language spatial representations that shape time conceptualization in real time. Further evidence for the implication of spatial neural system in the representation of time comes from neuropsychological data in which hemispatial neglect patients who ignore the left side of space, also have difficulties representing past events [33, 34].

In a recent meta-analysis [35], the automatic involvement of spatial information in the processing of words referring to time has been questioned (see [35]; for a meta-analysis). The argument was based on the finding that a strong and robust space-time congruency effect was found only when participants were asked to make an explicit judgment about the temporal content of verbal stimuli [12, 24, 30, 36]. In two studies designed to investigate that issue, Grasso et al. [23, 37] have shown however that robust space-time congruency effects can be observed even in an implicit temporal task involving conjugated verbs or pseudoverbs (i.e., lexical decision about whether a word is real or not) as long as participants answered with a lateral movement [23, 37]. In these studies, participants made lexical decisions to conjugated verbs or pseudoverbs and responded either by making spatially directed movements to the left or the right or by pressing a right or left response key. Space-time congruency effects occurred for laterally directed movements, but not for button presses, regardless of whether the movement was performed with a lateral swiping movement of the hand [23] or a lateral saccade with the eyes [37]. These results suggested that left- or rightward movements through space are critical for observing the space-time congruency effect when temporal processing is implicit and therefore provide strong empirical support for the automatic activation of the motor system in the very (spatial) representation of time-related words.

As concerns the spatial properties of the MTL, it has been suggested that the left-to-right MTL in western languages results from motor experience related to reading and writing. During writing, our hands, eyes, and attention move spatially from left to right, gradually shifting from what has been read/written—the past—to what will be read/written -the future- [25, 38, 39]. Empirically, this hypothesis is based on a large amount of cross-cultural data showing that

the direction of the space-time congruency effect follows the direction of the participants' writing system [25, 36]. In cultures that read and write from left to right, participants are faster to respond to past tense items with their left hand and future tense items with their right hand [12, 21, 24, 28, 40, 41], whereas the reverse pattern is found for participants whose writing system operates from right-to-left (e.g., Hebrew and Arabic scripts; [25, 36]). Casasanto and Bottini [38] were the first to directly assess the causal influence of reading expertise on the space-time congruency effect. In that study, Dutch participants had to judge the temporal content of sentences written either in a standard direction or in a mirror-reversed direction. Results revealed that the direction of space-time associations followed the direction of eye and hand movements: that is, left-past and right-future associations were observed when participants had to read sentences in the standard left-to-right direction, and right-past and left-future associations were found when participants had to read sentences in the right-to-left direction (for another example, see; [39]). These space-time congruency effects suggest that reading experience plays a role in the direction in which time is spatially represented.

If the abstract concept of time is grounded in directional movements performed during reading and writing, then the space-time congruency effect should correlate positively with the level of reading and writing expertise of the participants: more experienced readers should be more sensitive to space-time incongruities. Recently, the link between reading expertise and the processing of temporal order (i.e., before vs. after) was investigated in a developmental study using non-verbal stimuli [42]. Participants, ranging from preschoolers to adults, had to represent the temporal order of coloured circles presented sequentially on a computer by placing coloured cards in front of them [42]. Results revealed a left-to-right bias in organizing the order of events, even in pre-schoolers, but, more important, this bias increased with reading expertise of the children. Nevertheless, the claim that the scope of reading and writing experience correlates with the strength of the spatial representation of *words* related to the past and future still lacks direct empirical evidence.

Based on these theoretical and empirical considerations, the present study was designed to investigate the role of motor function and reading expertise in the space-time congruency effect that is observed during lexical processing of the abstract concept of time (i.e., past and future related words). One way to investigate the role of motor function in the space-time congruency effect is to manipulate whether congruency occurs at the perceptual (visual) or motor level. This was done in the study of Santiago et al. [12] who asked participants to categorize past and future words presented either on the left or right side of the screen, and to make lateralized keyboard responses using their left or right hand. Two types of space-time congruency were therefore manipulated. First, the congruency between the temporal content of the stimuli (past/future) and the left/right spatial location of the stimuli on the screen (i.e., perceptual congruency on screen), which is independent of motor action. Second, the congruency between the temporal content of the stimuli and the left/right spatial location of responses on the keyboard, which depends on motor action (i.e., motor congruency on the keyboard). Participants performed the same temporal categorisation twice: once with congruent key responses, and once with incongruent key responses in a counterbalanced order. Santiago et al. [12] found space-time congruency effects in both the perceptual and motor conditions, suggesting that incongruency on either the screen or the keyboard can interfere with performance. Yet, if activation of the motor system is critical for the occurrence of a space-time congruency effect during visual word processing, the effect should be stronger when the motor system is involved (the left/right localisation of response keys), rather than the visual system only (the left/right localisation of word stimuli on the screen). We tested this hypothesis in a replication of Santiago et al.'s experiment, with a larger and more heterogenous group of participants, where we directly compared the effect induced by the spatial manipulation of the keyboard to the effect

induced by the position of the stimuli on the screen. This comparison allows us to determine whether the space-time congruency effect is preferentially driven by motor or perceptual processing.

In addition, if the directionality of the MTL (future to the right and past to the left) were related to the directionality of reading and writing, reading expertise should correlate positively with the size of the space-time congruency effect. To test this second hypothesis, we included a standardized reading proficiency test, the SLS-Berlin [43]. This computerized reading test measures both reading comprehension and reading speed of sentences. It has been shown that the SLS-Berlin is an excellent tool for measuring interindividual differences in reading expertise [43]. As a linguistic control, we assessed another measure of language proficiency, vocabulary size, using a standardized test, the Lextale_FR [44]. By contrast to reading expertise, we predicted that vocabulary size should not influence the space-time congruency effect since it does not measure reading expertise per se.

In sum, we used an explicit temporal categorization task [12] to investigate the role of the motor system and of reading expertise on the space-time congruency effect. We expected to replicate Santiago et al.'s [12] finding that space-time congruency effects would be seen in both the screen and the keyboard conditions. However, since recent work suggests that the motor system plays a crucial role in lexical processing of the temporal content of words [e.g., 23], we predicted a stronger space-time congruency effect size in the keyboard-incongruent condition as compared to the screen-incongruent condition. Finally, we predicted a positive correlation between the size of the space-time congruency effect and reading expertise but not vocabulary size or age.

## Materials and methods

### Participants

Since one of the main objectives of the present study was to assess the link between reading expertise and the space-time congruency effect, we collected data from a wide range of ages in the general population rather than a restricted sample of undergraduate psychology students. This was done to increase inter-individual differences in reading expertise. As this was an online experiment on the Prolific.ac website, anyone could participate as long as they fulfilled certain criteria: i) to be aged at least 18 years old ii) to be a native French speaker and iii) have no cognitive disorders or deficits that could influence their performance (e.g., dyslexia). As this experiment took place during the COVID-19 pandemic, it was conducted entirely online between March and June 2021. Data collection was totally anonymous. Participants were informed that they were free to participate and to withdraw from the experiment at any time. Thus, ninety-one adults ranging in age from 18 to 72 years old participated in the experiment. Four individuals were not included in the final analyses due to high error rate (i.e., >25%). Four others had abnormal or aberrant RT distributions (many very fast or slow responses, or huge variability) and the data from these participants were excluded from the final analysis. The remaining 83 participants (36 women, 70 right-handed) were all French native speakers, reported normal or corrected-to-normal vision and ranged in age from 18 to 72 years old ($M$ = 30.69; $SD$ = 10.97). They all were recruited online, through the Prolific.ac website and received 10€/hours for their participation. The study was approved by the Institutional Review Board of Aix-Marseille University.

### Design and stimuli

We selected 84 past tense conjugated verbs and 84 future tense conjugated verbs from the Lexique 3 database [45]. Past and future words were matched with respect to length (7.92 versus

7.99 letters, *p* = .65, for past versus future words, respectively) and frequency (19.27 versus 8.42 occurrences per million, *p* = .11, for past versus future words, respectively). We used a Latin-Square design for creating 4 lists of 168 stimuli each (84 past words and 84 future words). Half of the past- or future- words could appear on the left of the screen and the other half on the right. For each session, participants saw one of the four lists, with each word presented once only and in random order.

## Apparatus

The experiment was programmed in PHP, JAVASCRIPT, and HTML, and was made available online. All participants performed the online experiment on their own personal computer and were asked to respond with their keyboard by pressing the "S" key for left-responses with their left-hand or the "L" key for right-responses with their right hand (AZERTY keyboard). All words were presented in 30-point arial font in black on a white background. When opening the web link of the online experiment, each participant was assigned to one of the four lists. We collected latencies (i.e., time between the onset of the stimulus and participants' responses) measured with millisecond precision (see Fig 1).

## Procedure

The experiment consisted of a temporal categorization task (i.e., does the stimulus refer to the past or the future?). Detailed instructions were displayed when opening the web link of the

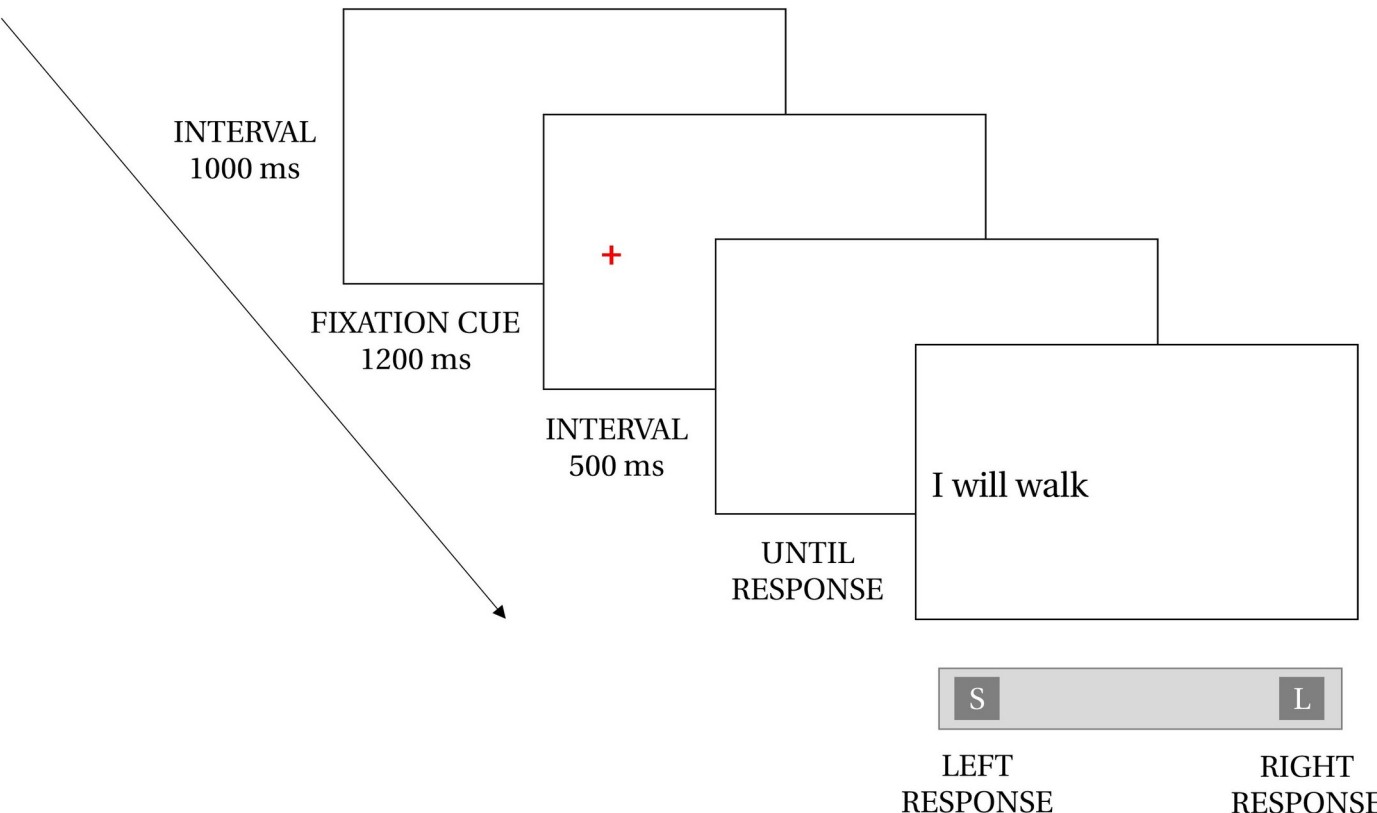

**Fig 1. Experimental design.** Words were displayed after a spatial fixation cue. Participants made their temporal categorization by pressing the left or the right response key (i.e., "S" and "L" key) on an AZERTY keyboard.

experiment. Participants were asked to perform the experiment in a quiet environment with no acoustic or visual distractions, and to start the task after completing a demographic questionnaire (i.e., age, gender, and dominant hand for writing). The experiment consisted of two independent sessions of 168 trials each, separated by 5 days and lasted for approximately 15 minutes. A participation link for the second session was sent to the participants 5 days after the first session. The tasks (temporal categorization) in both sessions were strictly identical except for the key response mapping, that is, in one session, the left-past and right-future response represented a space-time congruent key mapping, whereas in the other session the left-future and right-past response represented a space-time incongruent key mapping. The order of presentation of key response mapping was counterbalanced across participants. Instructions explained that words would appear either on the left or right side of the screen, which would always be preceded by a red cross to indicate where the word would appear. All participants were instructed to decide as rapidly and as accurately as possible whether the stimulus was a past- or future-word by pressing the left- or right-key of an AZERTY keyboard with their left and right hands, respectively. For each session, participants saw a total of 84 past-tense words and 84 future-tense words, presented in random order with half of each on the left of the screen and half on the right. As can be seen in Fig 1, at the beginning of each trial, a red fixation cross was displayed for 1200ms on the side where the word would appear, followed by a blank display for 500ms and then the stimulus (e.g., *I drew*) remained on the screen until the participants' response. Participants were asked to stop answering when they felt the need to take a break, and to resume when ready.

Note that independently of the type of stimuli presented, the combination of left-right stimulus locations on the screen together with left-right key responses could trigger the well-known Simon compatibility effect [46, 47]. When asked to produce a response at a location that is congruent with the spatial location of the stimulus on the screen (e.g., left key response for a left-sided stimulus), participants give faster responses than in incongruent trials (e.g., right key response for a left-sided stimulus). Although Santiago et al. [12] found no interaction between stimulus location on the screen and response location on the keyboard, which suggests the absence of a Simon effect, we added Simon congruency as a predictor in our statistical analyses to isolate space-time congruency from the (non-temporal) Simon effect. Altogether, the experiment therefore comprised four congruency conditions summarized as the A, B, C and D conditions, as depicted in Fig 2. All stimuli can be found online at https://osf.io/g2k6z/.

After the first session of the categorisation task, a link automatically redirected participants to the Lextale_FR [44] to assess vocabulary size. In this test, 120 words and pseudowords are presented in a fixed order and participants indicate whether they know the word or not. At the end of the categorisation task in the second session, a link automatically redirected participants to the French version of the SLS-Berlin [43] to assess reading expertise. This test consisted of the successive presentation of 77 semantically correct or incorrect sentences (e.g., "*Drunk drivers have a slower reaction time.*" or "*A rhinoceros is a wind instrument.*", respectively) at the centre of the screen, separated by a 1200ms blank screen and in a fixed order. Participants were asked to judge the plausibility or implausibility of the sentences by pressing a left or right key on their keyboard. The task stopped automatically after 3 minutes of sentence presentation (this duration did not include inter-stimulus intervals).

## Data analysis

We recorded and analyzed response accuracy (error rates, in percentage) and response latency (in milliseconds), that is the time interval between the onset of the stimulus and participants' keyboard response. Aberrant values corresponding to trials with latencies greater than 4000ms

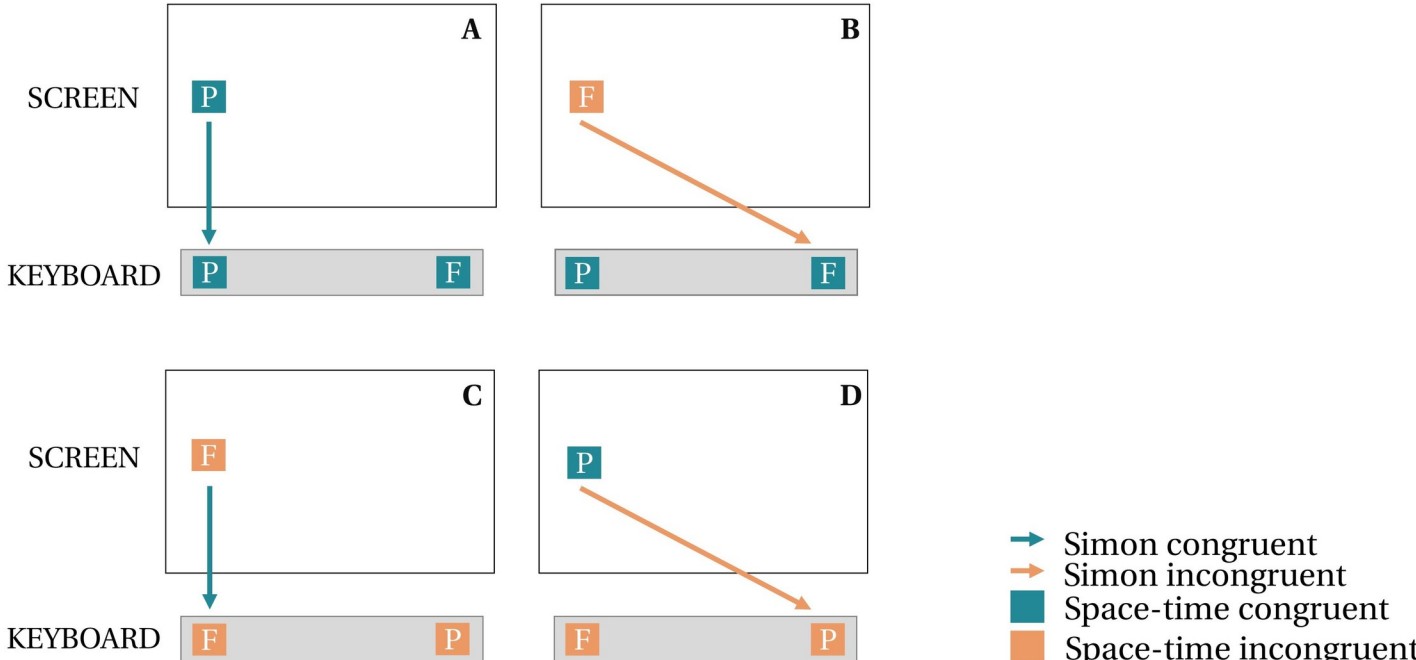

**Fig 2. Congruency conditions.** The figure illustrates the four congruency conditions for left-sided stimuli but note that four analogous conditions were also presented for right-sided stimuli. A and B correspond to keyboard congruent conditions, C and D correspond to keyboard incongruent conditions. In an orthogonal manner, A and D correspond to screen congruent conditions, B and C correspond to screen incongruent conditions. Finally, due to the orthogonalization of keyboard and screen location, A and C correspond to Simon congruent conditions and B and D correspond to Simon incongruent conditions. Note that Simon congruency was not manipulated orthogonally with space-time congruency effects but instead was simply a potential confound in the experimental design that we wished to control for, by including it as a predictor in the statistical analysis. P: past-tense stimulus. F: future-tense stimulus.

and less than 250ms (0.58% of trials) were discarded from the analyses. Response latencies were all inverse transformed (-1000 / RT) and the vocabulary and reading expertise data were standardized prior to the analyses.

Analyses of accuracy and latencies (for correct responses) were conducted with linear mixed-effects models [48] using the lmer() and glmer() functions from the lme4 package [49] in the R statistical computing environment [50]. We report unstandardized regression coefficients (*b*), standard errors (SEs), |*t*| or |*F*| values (for lmer), |*z*| values (for glmer), standardized mean difference (SMD, effect size), and corresponding p-values. For multilevel predictors (e.g., condition), we report F-values of the main effect, standard errors (SEs) and corresponding p-values computed with the anova() function from the lmerTest package [51]. We also report unstandardized regression coefficients (*b*), |Z| values and corresponding p-values for contrasts analyses computed with the pairwise() function from the emmeans package [52]. Regarding model building, we used the maximal random structure model that reached convergence [53]. The final model included by-participant and by-item random intercepts in all analyses that we report. In a forward stepwise model selection procedure, fixed and random effects of the models were selected according to the Akaike Information Criterion (AIC; [54]), the Bayesian Information Criterion (BIC; [55]), and the chi-squared log-likelihood ratio tests with regular maximum likelihood parameter estimation using the anova() function of the lmerTest package [51] for model comparison. Therefore, fixed effects, random effects, and random slopes were only included if they significantly improved the model's fit. Assumptions of the models were checked for each model using the check_model() function from the performance [56] package. Effect sizes for each predictor of the model were computed using the eff_size()

function of the emmeans package[52]. The eff_size() function computes a standardised mean difference, using pairwise differences of estimates divided by the (estimated) standard deviation of the population. To create figures, we used the emmeans package [52] that allowed us to compute the estimated marginals means (EMMs) for each model (see for instance Fig 4).

## Results

### Reaction times

The final model included screen congruency (congruent vs. incongruent), keyboard congruency (congruent vs. incongruent), Simon congruency (congruent vs. incongruent), and participant age as fixed effects, as well as standardized reading expertise (expressed by the sum of correct responses in the SLS-Berlin) and its interaction with each type of congruency, and by-participant and by-time intercepts. Note that no other predictor (e.g., vocabulary size) significantly improved the fit of the model. Results revealed a marginal effect of age ($b$ = -0.01, $SE$ = 0.01, $t$ = 1.79, $p$ = .076), a significant effect of reading expertise ($b$ = -0.01, $SE$ = 0.01, $t$ = 3.79, $p$ < .001), a significant Simon congruency effect ($b$ = -0.01, $SE$ = 0.01, $t$ = 3.05, $p$ < .001, SMD = 0.046), and a significant space-time congruency effect of key response mapping ($b$ = -0.01, $SE$ = 0.01, $t$ = 8.18, $p$ < .001, SMD = 0.131) (Fig 3). However, there was no significant space-time congruency effect of screen location ($b$ = -0.01, $SE$ = 0.01, $t$ = 1.28, $p$ = .259, SMD = 0.024). Interestingly, analyses revealed a significant interaction between reading expertise and space-time congruency of both key response mapping ($b$ = -0.01, $SE$ = 0.01, $t$ = 6.24, $p$ < .001) and screen location ($b$ = -0.01, $SE$ = 0.01, $t$ = 2.25, $p$ = .024), but no interaction with Simon congruency ($b$ = -0.01, $SE$ = 0.01, $t$ = 2.25, $p$ = .024). In other words, the better the reading expertise of the participants, the stronger the space-time congruency effect of the key response mapping and of the spatial location of the stimulus on screen (Fig 4).

### Error rate

No significant effect was observed on error rate (2.46% of the data).

## Discussion

The goal of the present study was two-fold. First, we wanted to investigate in an explicit temporal task using a large and relatively heterogeneous sample of participants whether the space-time congruency effect would be stronger if congruency involved a motor response (keyboard congruency) rather than perceptual input (screen congruency). Second, we wanted to investigate whether the nature and size of the space-time congruency effect can be linked to reading experience [25, 38, 39, 57].

To this end, we measured reading proficiency and the size of the space-time congruency effect using an adapted version of Santiago et al.'s paradigm [12]. This paradigm allowed us to manipulate two types of space-time congruency: 1) screen congruency (i.e., the congruency between the temporal content of stimuli (past vs. future) and their spatial location on screen (left vs. right), which is independent of motor action) and 2) keyboard congruency (i.e., the congruency between the temporal content of stimuli (past vs. future) and the associated key responses on the keyboard (left vs. right), which involves motor action). Combining lateralization of stimuli on the screen and lateralization of responses on the keyboard incidentally produced a Simon congruency condition, which was independent of the experimental manipulation of space-time congruency and was controlled for in the analyses. We predicted (1) space-time congruency effects both for screen congruency and keyboard congruency conditions similar to those found by Santiago et al. [12], (2) a larger space-time congruency effect

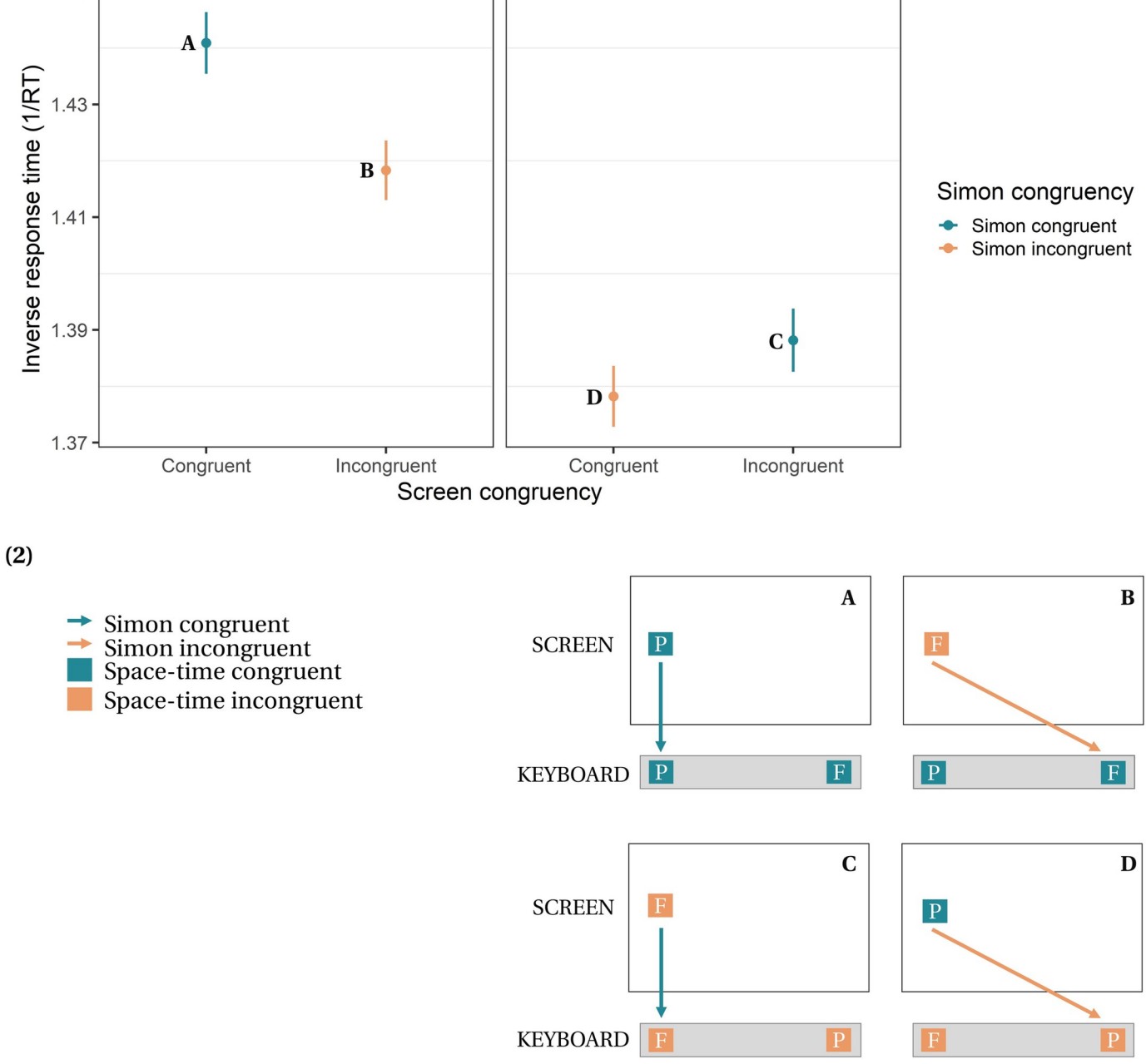

**Fig 3.** **(1)** Inverse response latencies (i.e., higher values correspond to faster responses) for all types of congruencies, as resumed in **(2)**. Error bars represent standard errors of the mean.

in the keyboard congruency condition, in which motor action is needed, compared to the screen congruency condition (i.e., cumulative effect), and (3) significant correlations between the size of the space-time congruency effect and participants' reading expertise.

Statistical analyses revealed a classic Simon congruency effect, which was independent of the two space-time congruency effects. All other things being equal, participants were slower for trials in which target location on the screen and response location on the keyboard were

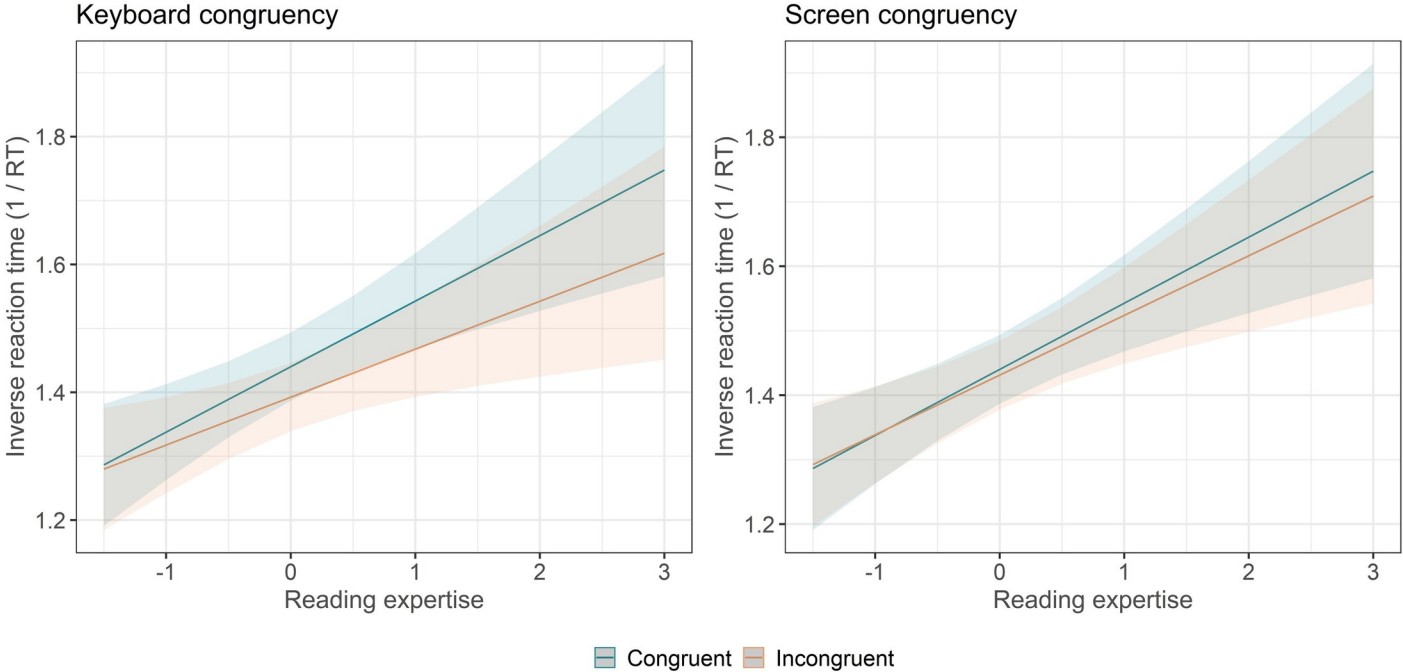

**Fig 4. Estimated values from the mixed models for inverse reaction times showing the interaction between reading expertise (i.e., standardised SLS score) and space-time congruencies (keyboard and screen congruency).** The greater the reading experience, the greater the congruency effect, in particular in the keyboard conditions (left panel).

opposed. Crucially, our results revealed a strong space-time congruency effect under the keyboard congruency condition. Average response times were significantly longer when keyboard responses-mappings contradicted the spatial organisation of time according to the MTL (i.e., left-future and right-past keyboard responses), independently of other congruency conditions and of the order in which the congruent or incongruent response-mappings were given. Such a dissociation is crucial, because the space-time congruency of the keyboard response is independent of the spatial location and the temporal content of the stimuli on the screen. Moreover, the congruency effect was stronger for the keyboard congruency condition (SMD = 0.131) as compared to the screen congruency condition (SMD = 0.025). In other words, the space-time congruency effect was stronger when it was implemented via a motor response (i.e., keyboard congruency). Together, these results can be taken to reinforce the hypothesis that the motor system is engaged when processing the temporal content or grammatical time of a word, which suggests that time is at least partially embodied in movement through space.

Before endorsing such an interpretation, one needs to consider alternative accounts. The most relevant alternative account is that space-time congruency effects in binary classification tasks could be explained by the polarity of the responses rather than the involvement of the motor system (see for exemple: [58]). According to this account, people assign either a positive or a negative polarity to both stimulus and response alternatives, which results in faster reaction times when the polarity of the stimulus and the polarity of the response match. According to the polarity framework, perceptual or conceptual overlaps between the spatial and temporal dimension of the stimulus or response would not be necessary to generate these congruency effects [58]. In our study, the fact that we observed a main effect of keyboard congruency (i.e., longer response times when the tense-keyboard response-mappings contradicted the left-to-

right MTL), whether or not the stimulus screen location was also compatible with the MTL, argues against the predictions of the polarity framework [see also Santiago and Lakens [59], who empirically tested the polarity principle and failed to support its predictions]. In our view, our results indicate that the abstract temporal concepts of past and future are at least partially represented spatially and particularly in the motor system.

Our second hypothesis concerned the role of reading expertise on the spatial direction of the MTL. As predicted, the effect of space-time keyboard congruency interacted with the level of reading expertise, with a higher level of reading expertise being associated with a stronger space-time congruency effect. Moreover, although the effect of space-time screen congruency was not significant in itself, it was also influenced by reading expertise: the higher the participants' level of reading expertise, the larger the size of the space-time screen congruency effect. By contrast, there was no relationship between the space-time congruency effect and vocabulary size, suggesting that the spatial representation of time does not reflect linguistic proficiency in general but is linked more specifically to reading habits. In addition, although participants' reading expertise correlated significantly with the size of both keyboard and screen space-time congruency effects, it did not correlate with the size of the Simon effect. As Simon congruency is fully independent of the lexical content of stimuli, this dissociation considerably strengthens the hypothesis that the direction of the spatial representation of time (i.e., the MTL) may derive from the direction of movement during reading and writing experience. Finally, conducting an online experiment allowed us to test a large sample of the population across a considerable age range. The final mixed model included participant age as a predictor, allowing us to estimate the effect of expertise while controlling for age. Interestingly, controlling for the effect of age did not modulate the effect of reading expertise and its interaction with space-time congruency effects. This suggests that the driving force behind this correlation is specific to reading experience rather than general life experience (chronological age).

Overall, these results replicate previous findings [20, 23, 37] and reinforce the hypothesis that the lexical processing of temporal stimuli is partially underpinned by motor, as well as spatial, processing systems. Moreover, we report a significant correlation between reading expertise and space-time congruency effects, which could account for the influence of people's cultural habits (i.e., direction of eye movements while reading) on their tendency to organize time spatially on a left-to-right MTL. This result is in line with the claim that the use of the spatial and motor systems for representing time results from reading and writing experience that creates an association between the movement of the eyes and hands rightwards or leftwards through space and the chronological order in which the left and right sides of space are experienced [38].

As for the neural mechanisms that associate the motor and spatial systems to the semantic representation of time, the neural reuse model [e.g., 60, 61] provides an interesting explanation. This model uses basic neurobiological insights about the formation of neural networks such as Hebbian learning–and, in particular, correlational learning–to explain how a set of sensory and motor neurons could be functionally involved in semantic processing. In short, the correlated patterns of neural activity that are present whenever a word or concept is experienced leads to the formation of strongly interconnected sets of cells distributed over the brain called action-perception circuits (APC). Regarding the abstract concept of time, it could be hypothesized that repeated experience of movements that start at one point in space and end later in time at another point associate spatial and temporal information together via correlated neural mechanisms. Repeated correlational patterns would ultimately constitute an APC consisting of, among other things, sensorimotor and spatial system that would be automatically activated during the processing of the concept of time. Considering the mechanisms proposed by the neural reuse model [e.g., 60, 61], we believe that abstracts temporal concepts can

be grounded in sensorimotor experience as concrete concepts. Our findings, together with those of previous studies [20, 23, 37], are compatible with the hypothesis that the abstract concept of time could be partially grounded in the temporal structure inherent to movement.

We conclude by endorsing Buonomano's [62] statement that "the nervous system of animals evolved in sophisticated ways to represent spatial coordinates, such as up and down, left and right, before it developed the ability to explicitly represent the temporal continuum of past, present and future. This line of reasoning is consistent with the theory that our ability to grasp the concept of time was coopted from the neural circuits that evolved to navigate, represent and understand space" [62, p. 182].

## Acknowledgments

We would like to thank Elisa Gavard for providing a French version of the SLS. and Boris Burle for his precious advice and comments.

## Author Contributions

**Conceptualization:** Camille L. Grasso, Johannes C. Ziegler, Marie Montant.

**Data curation:** Camille L. Grasso.

**Formal analysis:** Camille L. Grasso.

**Investigation:** Camille L. Grasso.

**Methodology:** Camille L. Grasso.

**Project administration:** Camille L. Grasso.

**Software:** Camille L. Grasso.

**Visualization:** Camille L. Grasso.

**Writing – original draft:** Camille L. Grasso.

**Writing – review & editing:** Camille L. Grasso, Johannes C. Ziegler, Jennifer T. Coull, Marie Montant.

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
