## [Decision Letter · Decision Letter 0]

21 Apr 2022

PONE-D-22-08528Embodied time: Effect of reading expertise on the spatial representation of past and futurePLOS ONE

Dear Dr. Grasso,

Thank you for submitting your manuscript to PLOS ONE. After careful consideration, we feel that it has merit but does not fully meet PLOS ONE’s publication criteria as it currently stands. Therefore, we invite you to submit a revised version of the manuscript that addresses the points raised during the review process.

As a further comment, I suggest to specify the period in which the data were acquired (before, during or after the pandemic).

We look forward to receiving your revised manuscript.

Kind regards,

Laura Barca, Ph.D

Academic Editor

PLOS ONE

Journal Requirements:

Reviewers' comments:

Reviewer's Responses to Questions

**Comments to the Author**

1. Is the manuscript technically sound, and do the data support the conclusions?

Reviewer #1: Partly

Reviewer #2: Yes

2. Has the statistical analysis been performed appropriately and rigorously? 

Reviewer #1: Yes

Reviewer #2: Yes

3. Have the authors made all data underlying the findings in their manuscript fully available?

Reviewer #1: Yes

Reviewer #2: Yes

4. Is the manuscript presented in an intelligible fashion and written in standard English?

Reviewer #1: Yes

Reviewer #2: Yes

5. Review Comments to the Author

Reviewer #1: The paper reports an experiment investigating the conceptualization of time in spatial terms. The authors present verbs referring to the past and the future either on the left or on the right side of the screen – past verbs were congruent when presented to the left and future verbs when presented to the right. They also manipulated the compatibility of the response – responses given by pressing a button on the left of the keyboard were congruent with past verbs, whereas pressing a button on the right with future verbs. Participants were asked to evaluate (via button press) whether the verbs were at the past or the future. Participants reading expertise was also collected. The authors report stronger incongruency effect for response compatibility (i.e., the side of response button) than for visuo-spatial compatibility (i.e., the side of the screen where the stimuli were presented). The size of this compatibility effect was associated with reading expertise (the higher the expertise, the larger the effect). Results are interpreted in the framework of embodied cognition.

The paper is well written and addresses a still interesting (although largely investigated) issue. At the same time, I have some theoretical concerns on the framing of the research and the interpretation of the results – both limited to the embodied perspective. Also, I struggled to fully understand the experimental design. All my concerns are detailed below. I hope they may help the authors in improving their work.

- Although I have nothing against the embodied interpretation of the reported effects, I think there is at least one main alternative interpretation that should be explicitly mentioned. Such interpretation has more simply to do with stimulus-stimulus and stimulus-response compatibility (for similar phenomena on up/down, small/big semantic properties, see, e.g., Santens & Gevers (2008), Cognition; Sellaro et al. (2015), Psychological Research; Treccani et al. (2019), Frontiers in Psychology). These phenomena would originate from task-related factors and may be read in the framework of polarity (e.g., Proctor & Xiong (2015), Current Direction in psychological Science). To my view, this interpretation is simpler, with one basic mechanism able to account for the pattern of data from many experiments. Although the authors may obviously keep the theoretical framework they prefer, I think they should introduce this alternative proposal at least to interpret their findings and also try to explain why their interpretation would be preferable. Similarly, some of the empirical findings investigating compatibility effects and not referring to the embodied perspective to frame their study should be (at least) cited.

- Throughout the manuscript the authors use different labels to refer to (likely) the same thing. In particular, when referring to motor action, they use different terms, as, e.g., motor execution and motor planning. However, different terms may refer to different things – to keep the example, motor planning is a higher-order level of processing that starts earlier than motor execution and in which the to-be-executed motor commands are selected and set up. I think the authors should be consistent in the terminology adopted and precisely clarify to what they refer to.

- Throughout the manuscript the authors (overtly and/or covertly) seem to propose the equation motor response (i.e., manual responses) = motor network. However, I find this equation too speculative: a) what the authors call motor congruency seems to be a spatial congruency effect, exactly as the one represented by stimuli dislocation on the screen (on this issue, see also my point above). The motor system is in no way (directly or indirectly) involved in the experimental manipulation. It is simply another spatial manipulation, with responses given either on the left or right side of the keyboard.

Following on this line, the authors repeatedly state that their findings support that abstract temporal concepts are represented in both spatial and motor networks. Maybe I’m missing something, but I struggle to understand on what data these conclusions are based on. The authors report some spatial compatibility effects, with no measure or manipulation directly targeting the activity within the spatial and/or motor network. The data, instead, seem to me easily accountable for in the framework of the polarity principle (e.g., Proctor & Cho (2006), Psychological Bulletin; Proctor & Xiong (2015), Current Direction in psychological Science). Note that I’m not saying that the authors must embrace this interpretation, but simply that they should not over-interpret their data.

- Task

The adopted task is quite explicit. Why didn’t the author use a more indirect task (as, e.g., lexical decision), or at least one less focused on the temporal dimension (e.g., a semantic categorization on an irrelevant dimension)? Can the authors exclude any specific contribution of the task to the reported effect?

- Experimental Design

I struggled to figure out the exact experimental design, and at the end I’m not sure I came out with the correct one. If I correctly understand, the authors have 3 factors, which are: a) Spatial congruency (i.e., the congruency between the semantic of the verb (past/future) and the position of the stimulus on the screen (left/right)); b) action congruency (i.e., the congruency between the semantic of the verb (past/future) and the side of the keyboard to release the response (left/right); c) Simon congruency (i.e., the congruency between the side of the response and the position of the stimulus on the screen (e.g., a verb at the future presented on the left side of the screen when the response is given pressing a button with the left hand on the left side of the keyboard)).

I have tried to understand all the possible combinations of the factors to understand what items belong to each cell of the design, but I couldn’t figure out the content of some cells. In particular, considering the Simon-congruent condition, what trials would be in the cell Spatial incongruent AND Action congruent, and in the cell Spatial congruent AND action incongruent? Similarly, for the Simon-incongruent condition, what trials would be in the cell Spatial congruent AND Action congruent, and in the cell Spatial incongruent AND action incongruent?

Is the design well balanced? Please clarify, also perhaps by adding a table with an example for each cell of the design.

Further, it is not clear to me why the authors did not try to disentangle the different compatibility effects by manipulating one at the time (and perhaps running two experiments, one per manipulation, with the same participants).

Minor points

- What is the distribution of reading expertise? Are there also participants with relatively low level of expertise or the variable has a skewed distribution toward the relatively high expertise?

- p. 14, ls. 357-358, check the statistics reported in parentheses.

- At the osf link, I couldn’t find the script of the analyses.

Reviewer #2: The study replicates in French a previous study by Santiago et al., investigating the effects of reading expertise on performance. The authors compare two types of space-time congruency effects, one visuo-spatial (screen congruency) and one motor (keyboard congruency), and show stronger effects for the motor incongruency than the visuo-spatial one. The size of the effect is predicted by reading expertise.

Overall comment

The study is well-conducted, the method is sound, and the writing is clear. I have some minor observations, mainly aimed at improving the quality of the study. I will outline them below.

Two general considerations: a) It would be useful to say from the start that the study is a replication one; b) it would be helpful to clarify from the start (and in the abstract) that the task consists of a temporal categorization one.

Page 3. When discussing abstract concepts, the authors might want to refer to authors who classified them as challenging for embodied cognition views, e.g., Borghi et al., 2017, The challenge of abstract concepts; G. Dove, 2021, The challenges of abstract concepts.

Page 3, line 75., The authors might want to refer more extensively to the seminal work of Lera Boroditsky, quite important in this area.

Line 97-on. When describing the procedure of the experiment by Grasso et al., please clarify that the words refer to time.

Line 129 (minor). Please substitute Casasanto et al. with Casasanto and colleagues.

Line 196. I think the authors should tell us more about how they selected the sample and determined the sample size.

Sample. In the sample, gender is unbalanced: might this have affected the results? How about education level? The authors could add some considerations on how demographic elements might have influenced the results.

Line 205. Technically, the Helsinki declaration would require the work to be preregistered, and this experiment is not.

Conclusion – In the conclusion, I think the authors should return to what was introduced at the beginning of the paper, i.e., the implications of their work for studies on abstract concepts.

6. PLOS authors have the option to publish the peer review history of their article (what does this mean?). If published, this will include your full peer review and any attached files.

Reviewer #1: No

Reviewer #2: No

---

## [Author Response · Author response to Decision Letter 0]

4 Jul 2022

Response to editorial revision requests (in blue)

Dear Editor, 

We are very grateful to the editor and reviewers for their suggestions and comments which substantially helped Improve our manuscript entitled “Embodied time: Effect of reading expertise on the spatial representation of past and future”. Please find below a summary of how we addressed each point. 

Reviewers' comments:

Reviewer #1: 

1. Although I have nothing against the embodied interpretation of the reported effects, I think there is at least one main alternative interpretation that should be explicitly mentioned. Such interpretation has more simply to do with stimulus-stimulus and stimulus-response compatibility (for similar phenomena on up/down, small/big semantic properties, see, e.g., Santens & Gevers (2008), Cognition; Sellaro et al. (2015), Psychological Research; Treccani et al. (2019), Frontiers in Psychology). These phenomena would originate from task-related factors and may be read in the framework of polarity (e.g., Proctor & Xiong (2015), Current Direction in psychological Science). To my view, this interpretation is simpler, with one basic mechanism able to account for the pattern of data from many experiments. Although the authors may obviously keep the theoretical framework they prefer, I think they should introduce this alternative proposal at least to interpret their findings and also try to explain why their interpretation would be preferable. Similarly, some of the empirical findings investigating compatibility effects and not referring to the embodied perspective to frame their study should be (at least) cited.

RESPONSE: We thank the reviewer for this suggestion. We added the references to the polarity account and a discussion of our effects in terms of polarity. It should be noted, however, that Santiago and Lakens (2015) explicitly tested the predictions of the polarity hypothesis in a series of four experiments. Their Experiment 4 was a past-future categorization task, just like ours, and their results did not support the predictions in terms of polarity (see also Santiago, 2007). Nevertheless, as suggested, we added a discussion of our results in terms of the polarity account (see p.471-501).

2. Throughout the manuscript the authors use different labels to refer to (likely) the same thing. In particular, when referring to motor action, they use different terms, as, e.g., motor execution and motor planning. However, different terms may refer to different things – to keep the example, motor planning is a higher-order level of processing that starts earlier than motor execution and in which the to-be-executed motor commands are selected and set up. I think the authors should be consistent in the terminology adopted and precisely clarify to what they refer to.

RESPONSE: We thank the reviewer for spotting this inconsistency. We have clarified the use of our terminology and now use motor action consistently throughout the manuscript. 

3. Throughout the manuscript the authors (overtly and/or covertly) seem to propose the equation motor response (i.e., manual responses) = motor network. However, I find this equation too speculative: a) what the authors call motor congruency seems to be a spatial congruency effect, exactly as the one represented by stimuli dislocation on the screen (on this issue, see also my point above). The motor system is in no way (directly or indirectly) involved in the experimental manipulation. It is simply another spatial manipulation, with responses given either on the left or right side of the keyboard. Following on this line, the authors repeatedly state that their findings support that abstract temporal concepts are represented in both spatial and motor networks. Maybe I’m missing something, but I struggle to understand on what data these conclusions are based on. The authors report some spatial compatibility effects, with no measure or manipulation directly targeting the activity within the spatial and/or motor network. The data, instead, seem to me easily accountable for in the framework of the polarity principle (e.g., Proctor & Cho (2006), Psychological Bulletin; Proctor & Xiong (2015), Current Direction in psychological Science). Note that I’m not saying that the authors must embrace this interpretation, but simply that they should not over-interpret their data.

RESPONSE:

We actually addressed this very point in our previous study, published In JEP:LMC (Grasso et al. 2021), where we contrasted spatial compatibility effects without movement (button presses) and spatial compatibility effects with movement. The MTL effects were present only when the decision involved movement, which is why we referred to motor networks in reference to our previous work. Note also that the contrast between static responses (without movement, button presses) and dynamic responses (with movement) is actually a test of the polarity hypothesis. 

However, we agree that the terms used in the current manuscript are potentially ambiguous. We have now replaced motor network by motor system. In addition, we would like to clarify that we manipulated two types of space-time congruency in our experiment: i) the congruency between the temporal content of the stimuli and their spatial location on the screen, which we labelled 'spatial congruency' because this manipulation is not dependant on the side of keyboard responses, and ii) the space-time congruency of the response hands on the keyboard, independently of the spatial location of the stimuli on the screen, which we labelled 'keyboard congruency’. Note also that the congruency effect was stronger when the space-time congruency condition was induced by motor responses rather than by the location of the stimuli on the screen. This finding was taken to suggest that the localisation of the motor response (and not the spatial localisation of the stimuli) induced the congruency effect. We now explain this point more clearly in the manuscript. 

4. Task

The adopted task is quite explicit. Why didn’t the author use a more indirect task (as, e.g., lexical decision), or at least one less focused on the temporal dimension (e.g., a semantic categorization on an irrelevant dimension)? Can the authors exclude any specific contribution of the task to the reported effect?

RESPONSE: 

In our previous work published in JEP:LMC (Grasso et al., 2021), we did in fact use an implicit task (i.e., lexical decision) and we showed that dynamic motion played a key role in the emergence of the MTL effects. By contrast, the main goal of the present study was to determine the relative contribution of motor (keyboard compatibiility) and visual (screen compatibility) processing on the emergence of MTL congruency effects in an explicit temporal task (see introduction).

5. - Experimental Design

I struggled to figure out the exact experimental design, and at the end I’m not sure I came out with the correct one. If I correctly understand, the authors have 3 factors, which are: a) Spatial congruency (i.e., the congruency between the semantic of the verb (past/future) and the position of the stimulus on the screen (left/right)); b) action congruency (i.e., the congruency between the semantic of the verb (past/future) and the side of the keyboard to release the response (left/right); c) Simon congruency (i.e., the congruency between the side of the response and the position of the stimulus on the screen (e.g., a verb at the future presented on the left side of the screen when the response is given pressing a button with the left hand on the left side of the keyboard)).

I have tried to understand all the possible combinations of the factors to understand what items belong to each cell of the design, but I couldn’t figure out the content of some cells. In particular, considering the Simon-congruent condition, what trials would be in the cell Spatial incongruent AND Action congruent, and in the cell Spatial congruent AND action incongruent? Similarly, for the Simon-incongruent condition, what trials would be in the cell Spatial congruent AND Action congruent, and in the cell Spatial incongruent AND action incongruent?

Is the design well balanced? Please clarify, also perhaps by adding a table with an example for each cell of the design.

Further, it is not clear to me why the authors did not try to disentangle the different compatibility effects by manipulating one at the time (and perhaps running two experiments, one per manipulation, with the same participants).

RESPONSE: We agree with the reviewer that the experimental design is complex, and we have tried to make it clearer in the manuscript. The Simon congruency was not manipulated orthogonally but instead covaried with the other two congruency manipulations. Thus, we controlled for Simon congruency, but we did not manipulate it. All the experimental conditions are presented in the figure above. Note that we show conditions for the left side of space only. Conditions are strictly identical for the right side but are not shown for reasons of parsimony. We have added a sentence to explain this in the description of the figure.

To clarify, when keyboard responses are congruent (condition A and B), spatial congruency and Simon congruency are always confounded (congruent Simon = congruent spatial; incongruent Simon = incongruent spatial). Conversely, when keyboard responses are incongruent: Simon and spatial congruency oppose one another: if Simon is congruent, spatial screen location is incongruent and vice versa. 

- Simon-congruent, Spatial incongruent AND Action congruent: this is not a possible combination. 

- Simon-congruent, Spatial congruent AND action incongruent: condition D. 

- Simon-incongruent, Spatial congruent AND Action congruent: this is not a possible combination.

- Simon-incongruent, Spatial incongruent AND action incongruent: this is not a possible combination. 

Further, we voluntarily choose to manipulate keyboard congruency in two different sessions (within-participants) at an interval of two weeks For example, in the first session, a given participant used an MTL-congruent response mapping while in the second session the participant used an MTL-incongruent response mapping.

We tried to make all this point clearer in the method section. 

6. Minor points

- What is the distribution of reading expertise? Are there also participants with relatively low level of expertise or the variable has a skewed distribution toward the relatively high expertise?

RESPONSE: The SLS score is based on the number of sentences correctly read/judged in 3 minutes (max 77 sentences). We obtained a very good distribution of reading abilities ranging from 11.0 to 67.00 correctly judged sentences, with an average of 28.07. The distribution was somewhat skewed, with most participants having a medium level of expertise but a few having very high levels of expertise. However, a non-normal distribution of one of the predictor variables in a mixed-effect model is not a problem as long as the residuals of the fitted variable are normal, which is the case. Please find the distribution of SLS scores (figure1) and plots of the model assumptions in figure 2. This is now acknowledged. 

Figure 1. Distribution of reading expertise (expressed by the SLS standardised score). SLS scores were standardized using the scale() function in R, which centers and scales the columns of a numerical matrix. Higher values reflect greater reading expertise.

Figure 2. Visual check of the various model assumptions (normality of residuals, normality of random effects, linear relationship, homogeneity of variance, multicollinearity), fitted with the check_model() function of the package performance (Lüdecke et al., 2021).

- p. 14, ls. 357-358, check the statistics reported in parentheses.

RESPONSE: we corrected the corresponding statistics. 

- At the osf link, I couldn’t find the script of the analyses.

RESPONSE: we thanks the reviewer for pointing out this mistake and have now added the scripts. of the statistical analyses. 

Reviewer #2: 

Two general considerations:

1. It would be useful to say from the start that the study is a replication one; 

2. b) it would be helpful to clarify from the start (and in the abstract) that the task consists of a temporal categorization one.

RESPONSE: done

- Page 3. When discussing abstract concepts, the authors might want to refer to authors who classified them as challenging for embodied cognition views, e.g., Borghi et al., 2017, The challenge of abstract concepts; G. Dove, 2021, The challenges of abstract concepts.

RESPONSE: done.

- Page 3, line 75., The authors might want to refer more extensively to the seminal work of Lera Boroditsky, quite important in this area.

RESPONSE: done.

- Line 97-on. When describing the procedure of the experiment by Grasso et al., please clarify that the words refer to time.

RESPONSE: done.

- Line 129 (minor). Please substitute Casasanto et al. with Casasanto and colleagues.

RESPONSE: done.

- Line 196. I think the authors should tell us more about how they selected the sample and determined the sample size.

RESPONSE: We added more information about the selection of the sample. As concerns the sample size, we initially aimed for 100 participants (final inclusion of 91), which is largely sufficient to detect an effect size of d=.3 (found in our previous experiment with an implicit task) with a power of over 80% (see Brysbaert, 2019) 

- Sample. In the sample, gender is unbalanced: might this have affected the results? How about education level? The authors could add some considerations on how demographic elements might have influenced the results.

RESPONSE: Almost half of the participants were women (83 participants, 36 women) and, to our knowledge, no available data seems to suggest a gender effect. Concerning the laterality of the participants, this variable was tested and had no effect. We had no information on the education level of the participants. 

Line 205. Technically, the Helsinki declaration would require the work to be preregistered, and this experiment is not.

RESPONSE: We thank the reviewer for pointing this, we removed this sentence. 

- Conclusion – In the conclusion, I think the authors should return to what was introduced at the beginning of the paper, i.e., the implications of their work for studies on abstract concepts.

RESPONSE: We thank the reviewer for this suggestion. We agree that this part was missing, so we have added a few sentences in discussion (see l. 587-591). 

REFERENCES 

Brysbaert, M. 2019 How Many Participants Do We Have to Include in Properly Powered Experiments? A Tutorial of Power Analysis with Reference Tables. Journal of Cognition, 2(1): 16, pp. 1–38. DOI: https://doi.org/10.5334/joc.7

Lüdecke et al., (2021). performance: An R Package for Assessment, Comparison and Testing of Statistical Models. Journal of Open Source Software, 6(60), 3139. https://doi.org/10.21105/joss.03139

---

## [Decision Letter · Decision Letter 1]

20 Jul 2022

PONE-D-22-08528R1Embodied time: Effect of reading expertise on the spatial representation of past and futurePLOS ONE

Dear Dr. Grasso,

Thank you for submitting your revised manuscript to PLOS ONE.I have sent the revised manuscript to the initial reviewers, who both appreciated the changes made.However, one reviewer believes that some of his/her points have not been completely resolved, and require further work.

Therefore, we invite you to submit a revised version of the manuscript that addresses the points raised during the review process.

We look forward to receiving your revised manuscript.

Kind regards,

Laura Barca, Ph.D

Academic Editor

PLOS ONE

Journal Requirements:

Reviewers' comments:

Reviewer's Responses to Questions

**Comments to the Author**

1. If the authors have adequately addressed your comments raised in a previous round of review and you feel that this manuscript is now acceptable for publication, you may indicate that here to bypass the “Comments to the Author” section, enter your conflict of interest statement in the “Confidential to Editor” section, and submit your "Accept" recommendation.

Reviewer #1: (No Response)

Reviewer #2: All comments have been addressed

2. Is the manuscript technically sound, and do the data support the conclusions?

Reviewer #1: Partly

Reviewer #2: Yes

3. Has the statistical analysis been performed appropriately and rigorously? 

Reviewer #1: Yes

Reviewer #2: Yes

4. Have the authors made all data underlying the findings in their manuscript fully available?

Reviewer #1: Yes

Reviewer #2: Yes

5. Is the manuscript presented in an intelligible fashion and written in standard English?

Reviewer #1: Yes

Reviewer #2: Yes

6. Review Comments to the Author

Reviewer #1: I'm one of the reviewers of the previous round. I think the authors have done a good job in addressing most of my criticisms. I still have two concerns.

The first one is related to the experimental design. The authors have now better clarified their design. If I'm correct, in half cases (i.e., when keyboard responses are congruent) Simon condition is confused with the spatial condition, whereas in the other half cases (i.e., when keyboard responses are incongruent), Simon and spatial conditions are distinguishable. If this is the case, don't the authors think that this is problematic in terms of interpreting their findings? (In half trials, two types of congruency/incongruency are jointly at work, whereas in the other half, only one at time). Take, for example, figure 3: It shows the Simon congruency effect for the keyboard congruent and the keyboard incongruent condition. However, for the keyboard congruent condition, is it the Simon or the spatial condition (or both)? Although there is no main effect of space-time congruency of screen location, can we be sure that such variable does not exert any role when co-occur with the Simon condition?

The second one is related to some of my previous comments. In particular, I raised doubts about the need to bring up the motor system to explain the reported findings. The authors replied that an “abstract compatibility” interpretation does not hold and they already published a study proving the involvement of the motor system. The authors have (almost) convinced me on the goodness of their interpretation (although I still find it not very economic), but I would see a bit more caution on the way they embrace it. I still don't see strong proof in favor of the involvement of the motor system. The authors refer to their previous study (Grasso et al., 2021), which I decided to read to have a more complete view of their work. Although that study has an experiment directly manipulating the involvement of the motor system (Experiment 2), I have found the results of such experiment not fully convincing. In fact, in the ANOVA analysis, they report a strong two-way interaction between time and side (i.e., the congruency effect), but a weak three way interaction between time, side and device (which should be the critical result, but it is significant by participants, but not by items). This suggests that the device does not play a so strong role in modulating the effect. The three-way interaction becomes fully significant only when the two conditions requiring movements (trackpad and mouse) are collapsed and jointly compared against the keyboard condition (which does not require movement). However, it seems to me that there are no post-hoc comparisons, which makes hard to fully appreciate the pattern. This is also what happens in the mixed-effects model analysis. Also, looking at Figure 2, it seems that keyboard and mouse behave similarly, with the former showing a smaller (but not negligible) effect. To make a long story short, it seems to me that, although the motor component seems to play a role, it does not seem to be necessary to make the effect arise. I would suggest to slightly weaken some of the theoretical statements on the role of motor component.

Reviewer #2: (No Response)

7. PLOS authors have the option to publish the peer review history of their article (what does this mean?). If published, this will include your full peer review and any attached files.

Reviewer #1: No

Reviewer #2: No

---

## [Author Response · Author response to Decision Letter 1]

28 Sep 2022

The first one is related to the experimental design. The authors have now better clarified their design. If I'm correct, in half cases (i.e., when keyboard responses are congruent) Simon condition is confused with the spatial condition, whereas in the other half cases (i.e., when keyboard responses are incongruent), Simon and spatial conditions are distinguishable. If this is the case, don't the authors think that this is problematic in terms of interpreting their findings? (In half trials, two types of congruency/incongruency are jointly at work, whereas in the other half, only one at time). Take, for example, figure 3: It shows the Simon congruency effect for the keyboard congruent and the keyboard incongruent condition. 

RESPONSE: Yes, this is a fundamental limitation of the original Santiago et al. design used in the present study. In fact, we had addressed this confound in our previous research (Grasso et al, 2021) by presenting words in the center of the screen and asking participants to make left or right spatially directed movements. However, because our aim in the current study was to investigate both spatial congruency on the screen and spatial congruency of the keyboard (independently of one another) lateralization of both stimulus presentation and response hand had to be orthogonalised, which inevitably leads to a confound with the Simon effect. Unfortunately, there is no way to manipulate these three effects orthogonally. Thus, the best one can do is to control for the Simon effect by adding it to the statistical mixed effect model ( Model <- lmer(Data$ReverseRT ~ SLS_CORRECTE_s * Simon_Congruency + Keyboard_Rep * SLS_CORRECTE_s + Screen_Congruency * SLS_CORRECTE_s + Age + (1|New_ID)+(1|Item)) ) and being cautious in the interpretation of results. Including the Simon effect in the statistical model effectively partials out the part of the variance that is due to this variable. Also note that the analysis showed no interaction between screen congruency and Simon congruency and no interaction between keyboard congruency and Simon congruency (models were tested and compared using the anova() function of the lmerTest package for model comparison) . Thus, we are fairly confident that our results are not due to a confound with Simon. As requested, we mention this as a limitation of the design, and we interpret our results with caution. 

However, for the keyboard congruent condition, is it the Simon or the spatial condition (or both)? Although there is no main effect of space-time congruency of screen location, can we be sure that such variable does not exert any role when co-occur with the Simon condition?

RESPONSE: As we can see in the statistical analyses, there was no significant interaction between each type of congruency. Given that keyboard congruency did not interact with either screen or Simon congruency, any effect of screen or Simon congruency in just one of the keyboard conditions cannot be meaningfully interpreted. As a reminder, the keyboard congruency effect (congruent vs. incongruent) was completely orthogonal to Screen congruency and Simon congruency. Thus, regardless of Screen or Simon congruency/incongruency, RTs were slower in the incongruent keyboard condition (grouping conditions D and C) than in the congruent keyboard condition (grouping conditions A and B). This can be seen in the graph below.

The second one is related to some of my previous comments. In particular, I raised doubts about the need to bring up the motor system to explain the reported findings. The authors replied that an “abstract compatibility” interpretation does not hold and they already published a study proving the involvement of the motor system. The authors have (almost) convinced me on the goodness of their interpretation (although I still find it not very economic), but I would see a bit more caution on the way they embrace it. I still don't see strong proof in favor of the involvement of the motor system. The authors refer to their previous study (Grasso et al., 2021), which I decided to read to have a more complete view of their work. Although that study has an experiment directly manipulating the involvement of the motor system (Experiment 2), I have found the results of such experiment not fully convincing. In fact, in the ANOVA analysis, they report a strong three-way interaction between time and side (i.e., the congruency effect), but a weak three-way interaction between time, side and device (which should be the critical result, but it is significant by participants, but not by items). This suggests that the device does not play a so strong role in modulating the effect. The three-way interaction becomes fully significant only when the two conditions requiring movements (trackpad and mouse) are collapsed and jointly compared against the keyboard condition (which does not require movement). However, it seems to me that there are no post-hoc comparisons, which makes hard to fully appreciate the pattern. This is also what happens in the mixed-effects model analysis. Also, looking at Figure 2, it seems that keyboard and mouse behave similarly, with the former showing a smaller (but not negligible) effect. To make a long story short, it seems to me that, although the motor component seems to play a role, it does not seem to be necessary to make the effect arise. I would suggest to slightly weaken some of the theoretical statements on the role of motor component. 

RESPONSE: We agree with the reviewer that there must be more to the effect than the involvement of spatially directed movements (motor system). Otherwise, no effect would be observed in explicit timing tasks with static keypress responses (see for example the meta-analyses von Sobbe et al. 2019). It is extremely difficult to obtain significant triple interactions by subjects and items and you correctly point out that the triple interaction in the ANOVA analyses was only significant by subjects and items when we collapsed the two movement conditions together (Note, however, that degrees of freedom for items were quite low, 79). Nevertheless, when subjects and items were used as random variables in the equivalent mixed effect model (which has become the standard for analyzing psycholinguistic experiments), the triple interaction was significant. Nevertheless, we decided to follow your request and slightly weakened some of theoretical statements on the role of motor components (see l.45-46, l.427, l.442, l.466, l.486, and l.491).

---

## [Editor Report · Decision Letter 2]

4 Oct 2022

Embodied time: Effect of reading expertise on the spatial representation of past and future

PONE-D-22-08528R2

Dear Dr. Grasso,

We’re pleased to inform you that your manuscript has been judged scientifically suitable for publication and will be formally accepted for publication once it meets all outstanding technical requirements.

Kind regards,

Laura Barca, Ph.D

Academic Editor

PLOS ONE
---

## [Editor Report · Acceptance letter]

7 Oct 2022

PONE-D-22-08528R2 

Embodied time: Effect of reading expertise on the spatial representation of past and future 

Dear Dr. Grasso:

I'm pleased to inform you that your manuscript has been deemed suitable for publication in PLOS ONE. Congratulations! Your manuscript is now with our production department. 

Kind regards, 

on behalf of

Dr. Laura Barca 

Academic Editor

PLOS ONE